# Periodic temperature changes drive the proliferation of self-replicating RNAs in vesicle populations

Elia Salibi[1,3], Benedikt Peter[2,3], Petra Schwille [2] ✉ & Hannes Mutschler [1] ✉

Growth and division of biological cells are based on the complex orchestration of spatiotemporally controlled reactions driven by highly evolved proteins. In contrast, it remains unknown how their primordial predecessors could achieve a stable inheritance of cytosolic components before the advent of translation. An attractive scenario assumes that periodic changes of environmental conditions acted as pacemakers for the proliferation of early protocells. Using catalytic RNA (ribozymes) as models for primitive biocatalytic molecules, we demonstrate that the repeated freezing and thawing of aqueous solutions enables the assembly of active ribozymes from inactive precursors encapsulated in separate lipid vesicle populations. Furthermore, we show that encapsulated ribozyme replicators can overcome freezing-induced content loss and successive dilution by freeze-thaw driven propagation in feedstock vesicles. Thus, cyclic freezing and melting of aqueous solvents – a plausible physicochemical driver likely present on early Earth – provides a simple scenario that uncouples compartment growth and division from RNA self-replication, while maintaining the propagation of these replicators inside new vesicle populations.

Early cellular life on Earth is thought to have started with the encapsulation of self-replicating informational subsystems in primitive compartments[1]. Encapsulation of biomolecules can occur in a variety of microenvironments, such as adhesive mineral surfaces, fatty acid vesicles, liposomes, coacervates, and even peptide-based membranes[2]. The transfer of genetic information is primarily associated with template-dependent replication of nucleic acids[3,4], but can also be performed to a limited extent with peptides[5] and small organic molecules[6]. Compartmentalisation is required to isolate and protect the autocatalytically replicating information system from environmental perturbations and parasitic replicators and to establish phenotype-genotype coupling[7,8]. A popular model for protocells involves a "replicase", a catalytic nucleic acid that aids templated replication of nucleic acid polymers, which is encapsulated in a membrane vesicle[2]. While initial concepts of such vesicular protocells

evolved around fatty acids, which are accessible via prebiotically plausible synthesis pathways[9], their low stability and lack of tolerance towards physicochemical parameters such as pH and ionic strength made reconciling them with nucleic acid catalysis challenging[10,11]. Recent studies demonstrating the non-enzymatic synthesis of diacyl phospholipids either by the phosphorylation of diacyl glycerol[12], the selective alkylation of phosphoglycerol[13], or the alkylation of an acyl phospholipid precursor[14], strengthen the hypothesis that early protocells may have also had membranes containing simple phospholipids[15].

Vertical transfer of genetic information is a prerequisite for a species' ability to proliferate. Horizontal gene transfer, on the other hand, is known to be a powerful mechanism for intra- and interspecies dissemination of genetic innovation[16,17]. Whilst modern cells rely on protein-based machineries for replication of both compartments and genetic polymers as well as for the uptake of exogenous genetic

[1]Department of Chemistry and Chemical Biology, TU Dortmund University, Otto-Hahn-Str. 4a, 44227 Dortmund, Germany. [2]Department of Cellular and Molecular Biophysics, Max Planck Institute of Biochemistry, Am Klopferspitz 18, 82152 Martinsried, Germany. [3]These authors contributed equally: Elia Salibi, Benedikt Peter. ✉e-mail: schwille@biochem.mpg.de; hannes.mutschler@tu-dortmund.de

materials, such systems were not available during early stages of molecular evolution. A possible plausible alternative to the complex extant biological processes that mediate the exchange and spread of genetic molecules is achieved by repetitive physical and/or chemical processes altering compartment permeability, such as periodic pH changes, temperature oscillations or dehydration cycles[18–20]. In particular, the freezing of salted water leads to the formation of interstitial brines amongst the growing ice crystals, which concentrates dilute solutes and ions in solution[21]. For this reason, liquid brine phases in water-ice matrices are considered attractive, potentially prebiotic environmental niches because they favour reactions otherwise incompatible with dilute aqueous conditions. For example, UV-irradiation of urea solutions followed by freeze-thaw (FT) cycles enables prebiotic synthesis of purines and pyrimidines[22]. Frozen conditions have also been shown to promote both nucleotide activation and non-enzymatic templated copying of RNA[23], as well as host more complex ribozyme reactions that increase complexity of the starting RNA pools[24–26]. In addition, a plausible pathway for the synthesis of peptide-RNA is promoted by water-ice formation and the accompanying pH change[27]. FT cycles have also been shown to enable content exchange between membrane vesicles without requiring complex protein machineries. For instance, repeated liquid nitrogen-based freeze-thaw cycles of pelleted lipid vesicles enable the coupling of fusion and fission of liposomes with RNA replication by a modern viral RNA replicase, and the distribution of RNAs within the vesicle population[28]. Previously, our groups investigated the effects of temperature cycling on giant unilamellar vesicles (GUVs), showing that freezing at −80 °C followed by passive thawing allowed content exchange between densely packed GUVs independently from fusion and fission[29].

Here, we showcase that when a mixture of GUVs, encapsulating either an RNA substrate or a ribozyme, is subjected to a freeze-thaw cycle, the contents of both GUV populations mix during thawing enabling encapsulated RNA catalysis under unfrozen conditions. The system is independent from the type of ribozyme used and promotes both RNA cleavage and ligation activity. We further demonstrate that the use of an autocatalytic ligase ribozyme replicator enables concurrent ribozyme amplification and vesicle content exchange, thus leading to the proliferation of ribozyme-containing vesicles during serial transfer experiments, when combined with the feeding of substrate-encapsulating GUVs. Together, these experiments present a model for how early RNA-protocells may have been able to grow and proliferate in a plausible geochemical environment exhibiting cyclical freezing and thawing of aqueous solvents.

## Results

### Freeze-thawing allows content exchange and assembly of encapsulated ribozymes

Initially, we sought to explore whether it is possible to exchange RNA strands encapsulated in different GUV populations by freeze-thaw (FT) cycles, thereby triggering a ribozyme reaction (Fig. 1a). In these experiments, we opted for a well-characterised hammerhead ribozyme (HH-min) that cleaves an RNA substrate strand, which either contains only a fluorescent dye (Cy5) at the 5′-end (named HH-Sub) or a fluorescent dye at the 5′-end (FAM) and a black hole quencher (BHQ1) at the 3′-end (named HH-FQ-Sub)[30] (Supplementary Figs. 1a and 2). Cleavage of the RNA substrate can therefore be characterised both microscopically by the increase in fluorescence upon HH-FQ-Sub cleavage or by classical denaturing polyacrylamide gel electrophoresis (PAGE) upon HH-Sub cleavage. We encapsulated both HH-min and quenched HH-FQ-Sub in separate GUV populations, composed of 1-palmitoyl-2-oleoyl-phsophatidylcholine (POPC), before they were combined in a 1:1 ratio for each experiment. We performed a direct quantification of the intravesicular fluorescence of the different vesicle populations and observed a strong increase in fluorescence intensity in both GUV

populations after a single freeze-thaw (FT) cycle. This observation suggested that efficient RNA exchange had occurred between both vesicle populations during FT-cycling resulting in HH-min cleavage in both vesicle types (Fig. 1b). We could confirm that RNA cleavage was both HH-min and FT-dependent since samples that were either not subjected to FT-cycling or contained either no HH-min or a catalytically inactive HH variant (HH-mut[30], Supplementary Table 1) showed a ~7-fold lower fluorescence post-incubation compared to HH-min samples that were exposed to FT-cycles (Fig. 1c; Supplementary Fig. 3). To estimate the percentage of cleaved HH-FQ-Sub, we normalised the data obtained to a control experiment in which HH-min and HH-FQ-Sub were encapsulated in the same GUV population (Supplementary Fig. 4). We estimate that approximately 80% of HH-FQ-Sub was cleaved after a single FT cycle. Finally, we collected the pre- and post-cycling GUV solutions and analysed the resulting RNA species by polyacrylamide gel electrophoresis (PAGE) to verify that FT-induced content exchange led to specific HH-sub cleavage. As expected, almost 80% of the collected HH-sub RNA had undergone site-specific cleavage by HH-min, corroborating the microscopy data that fluorescence increase is indeed a result of RNA catalysis and FT-cycling (Fig. 1d). In contrast, omitting FT-cycling resulted in 20% cleavage. This residual activity was presumably caused by extravesicular cleavage by leaked HH-min and HH-sub that had occurred during sample preparation, e.g., through shear forces-induced rupturing of GUVs. Taken together, these data demonstrate that FT cycles do provide a means for the exchange of RNA between neighbouring membrane vesicles and can therefore trigger catalysis of otherwise inactive ribozymes.

### Assembly and self-replication of encapsulated ligase ribozymes

Next, we sought to model more prebiotically relevant ribozyme activities, namely, quasi-irreversible RNA ligation. Typically, RNA ligase ribozymes require higher magnesium concentrations than small nucleolytic ribozymes[31]. For this reason, it was first necessary to test our encapsulation protocol under higher $Mg^{2+}$ concentrations. An initial magnesium concentration screen on GUV preparation revealed that RNA encapsulation is efficient up to 30 mM $MgCl_2$, suggesting that the vesicles could host catalytic RNAs with higher magnesium requirements than the hammerhead ribozyme (Supplementary Fig. 5). To confirm this compatibility, we tested a variant of the R3C ligase ribozyme[32] (rt-F) that catalyses ligation of a quenched tetramethylrhodamine-labelled substrate (FQ-B) and short RNA strand (rt-A-short), resulting in the release of the fluorophore and an increase in green fluorescence[33] (Supplementary Figs. 1b, 6 and 7). Similar to the hammerhead system, we observed a strong increase in fluorescence intensity (1.8- and 4.4-fold for 2 and 10 μM substrate, respectively) under the same experimental setup and freeze-thaw (FT) regimen, indicating ribozyme-catalysed ligation. We noticed substantial heterogeneity in fluorescence intensity per GUV after thawing, suggesting non-homogeneous content exchange and/or post-mixing ligation. This effect was more pronounced in the R3C system compared the hammerhead system and likely resulted from the higher magnesium concentrations, which already introduced a concentration dependent variance of RNA encapsulation among vesicles (Supplementary Fig. 5). Additional factors contributing to the heterogeneity might stem from non-ideal mixing of initial GUV populations and therefore local variations in the intravesicular levels of the ribozyme and its substrates after thawing. Moreover, the observed RNA-membrane interactions for the FQ-B substrate (Supplementary Fig. 8), as well as local variations in content loss incurred upon freeze-thaw cycling (Supplementary Fig. 9) likely contribute to non-homogeneous intravesicular mixing.

Having shown that the R3C system is active under freeze-thaw (FT) cycling conditions, we further set out to explore if the same system can be leveraged to establish autocatalytic RNA self-replication inside vesicles. We aimed to model a potential prebiotic scenario addressing a fundamental feature of early protocells, namely the

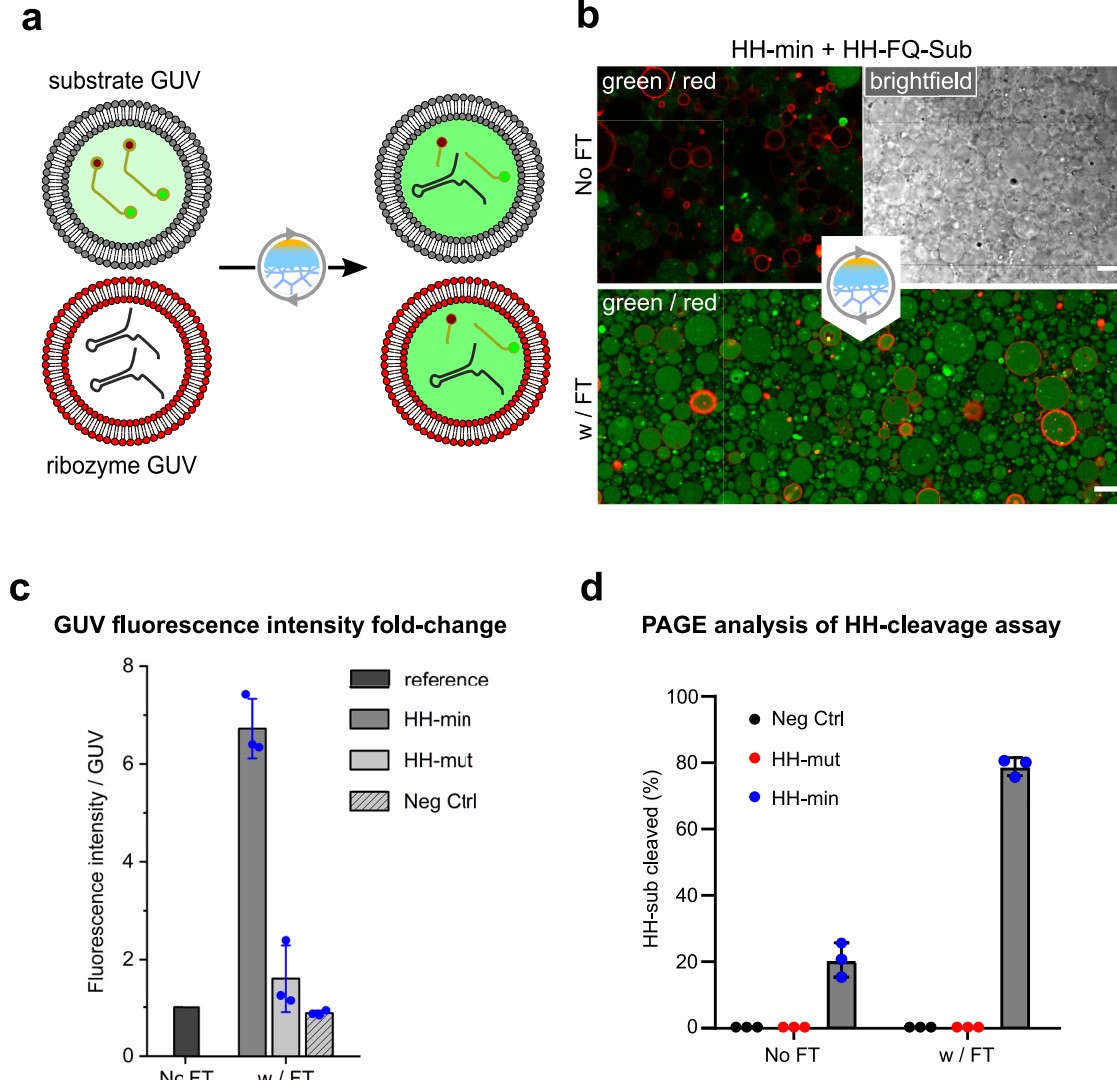

**Fig. 1 | Freeze-thaw (FT) driven catalysis of a split hammerhead system encapsulated in vesicles. a** Illustration showing reaction setup for the split hammerhead system encapsulated in giant unilamellar vesicles (GUVs). HH-min containing GUVs (red membrane lipid) were combined with HH-FQ-Sub containing GUVs (grey) in 1:1 ratio and subjected to a single freeze-thaw (FT) cycle. **b** Confocal microscopy of 2.5 μM HH-FQ-Sub GUVs and 5 μM HH-min GUVs before (top) and after (bottom) a FT-cycle in 20 mM Tris-HCl pH 8.3, 4 mM MgCl₂, and 900 mM sucrose/glucose. **c** Fold change of fluorescence intensity per GUV after one FT-cycle, normalised to the average fluorescence intensity of all GUVs in the sample before freezing. Negative Control (Neg Ctrl) had empty vesicles combined with substrate vesicles. Data are presented as box charts with mean values of GUV fluorescence +/− SD from $n = 3$ independent experiments. **d** Cleavage yields of 2.5 μM HH-Sub encapsulated in GUVs mixed with empty GUVs (Neg Ctrl - black), GUVs encapsulating 5 μM inactive HH-mut (red), or GUVs encapsulating 5 μM HH-min (blue) before and after subjecting them to a FT cycle followed by a 1-h incubation at 37 °C in 20 mM Tris-HCl pH 8.3, 4 mM MgCl₂, and 900 mM sucrose. Data are presented as box charts with mean values +/− SD from $n = 3$ independent experiments; individual data points are shown as dots. Scale bars represent 15 μm before FT-cycling and 10 μm after FT-cycling.

autocatalytic replication of genetic material and its horizontal transfer via a simple, environment-driven inter-protocellular exchange mechanism. To this end, we selected an optimised self-replicating R3C enzyme, which was previously obtained by directed evolution by Robertson and Joyce[4]. In this system, the ligase ribozyme (F1) replicates itself by joining two oligonucleotide substrates (Hyper-A and B, Supplementary Fig. 1c). A reporter substrate (Cy5-A) was designed to carry an inactivating single point mutation (G38A) and a Cy5 fluorophore at the 5'-end (Supplementary Table 1).

After characterising and establishing suitable self-replication activities of F1 under bulk conditions (Supplementary Figs. 10–15), we probed whether F1 self-replication could be triggered by FT-driven exchange of substrate containing GUVs with vesicles encapsulating the full-length ribozyme. Specifically, we aimed to explore if R3C-based replicators can "propagate" in vesicle populations through repeated

"infection" of substrate filled GUVs driven by freeze-thaw (FT) cycling (Fig. 2a). To this end, we mixed equal amounts of vesicles containing 1 μM or 2 μM F1 with vesicles containing the two substrates. To exclude Hyper-A based background ligation before content mixing, we separated the substrates Hyper-A (10 μM with 2.5 μM Cy5-A as reporter strand) and B (15 μM) in different GUV populations. As expected, no ligation occurred during co-incubation of the three populations at 42 °C. However, when the system was exposed to repeated FT-cycling, we observed increasing levels of F1 formation (Fig. 2b; Supplementary Fig. 16). After one FT cycle, between 4% (1 μM initial F1) and 10% (2 μM initial F1) of Cy5-A had been incorporated in full-length ribozymes. Further cycling of the same samples increased ligation yields to ~6% (1 μM initial F1) and ~13% (2 μM initial F1). We could show that the majority of the activity resulted from content mixing of F1 with both substrates, as GUV mixtures in which F1-containing GUVs had been

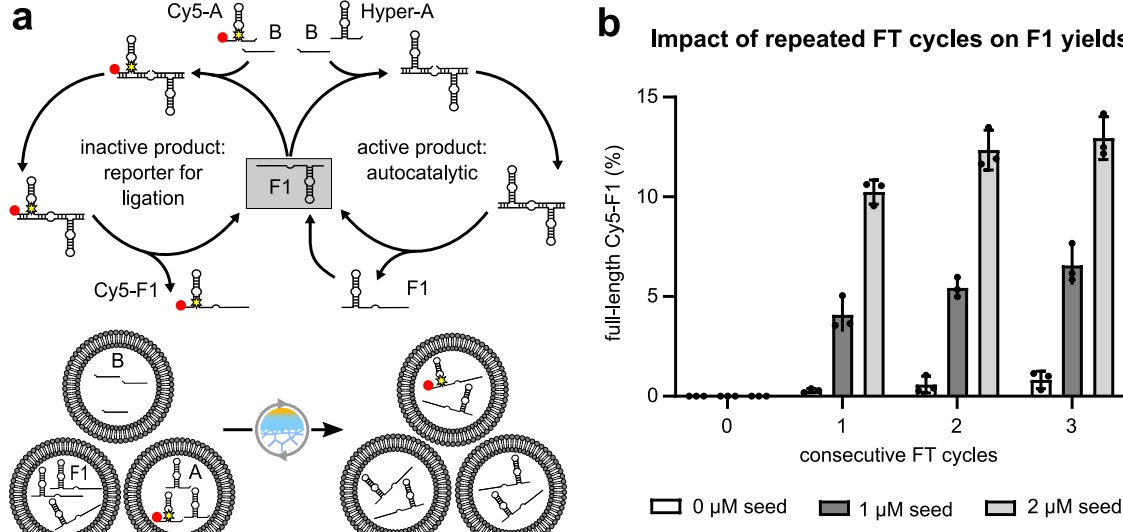

**Fig. 2 | Autocatalytic self-replication of GUV-encapsulated F1 ligase after FT-induced content mixing. a** The reaction cycle of the autocatalytic ligase system starting from an initial seed of unlabelled ribozyme (F1) (top panel) and the three distinct vesicle populations used for freeze-thaw (FT)-driven self-replication of RNAs (bottom panel). Addition of the fluorescently labelled substrate Cy5-A yields a catalytically inactive reporter product (Cy5-F1) that allows the monitoring of the reaction progress using fluorescent PAGE analysis. To make F1 self-replication dependent on GUV content exchange, separate GUV populations encapsulating either the F1 seed, substrate A or substrate B were mixed in a 1:1:1 ratio before FT

cycling. **b** Cy5-F1 ligation yields as a function of FT-cycles. GUVs encapsulating either 1 μM F1, 10 μM Hyper-A (with 2.5 μM Cy5-A) or 15 μM B substrate were combined, subjected to varying numbers of FT cycles, incubated for 1 h at 42 °C and subsequently analysed by PAGE. For negative controls, empty vesicles (0 μM seed) were combined with substrate vesicles in the same ratios and subjected to the same conditions. Modest increase of F1 during FT-cycling in the absence of F1 seed resulted from Hyper-A background activity (Supplementary Figs. 10–14). Data are presented as bar chart mean values +/− SD of the 3 independent experiments shown.

replaced with empty vesicles showed less than 1% of F1 background formation under these conditions even after three FT cycles (Fig. 2b, Neg Ctrl). The observed correlation between the number of FT cycles and ribozyme concentration in systems with F1 containing GUVs suggested that repeated cycles enhanced content exchange between the different populations and increased the total fraction of ligated product despite the concurrent content loss that occurred during transient membrane disruption and vesicle fragmentation (Supplementary Figs. 9 and 17). The freeze-thaw cycles also result in a temporal decoupling of RNA propagation and actual self-replication: since R3C ligase is nearly inactive under both frozen conditions and during thawing (Supplementary Fig. 18), actual self-replication occurs predominantly under encapsulated aqueous conditions.

### Sustained self-replication of encapsulated ligase ribozymes

After identifying suitable RNA concentrations and the optimal freeze-thaw (FT) cycling protocol (10-fold excess substrates; three FT cycles followed by an hour at 42 °C), we aimed to explore whether R3C-based replicators can persist in vesicle populations during serial transfer experiments by escaping selective pressure in an open system through repeated "infection" of substrate-filled GUVs driven by FT-cycling. Specifically, we set out to leverage the combination of FT cycle-induced content mixing and autocatalytic replication to simulate survival of a ribozyme under serial transfer conditions, i.e., when vesicles containing self-replicators are challenged with ongoing dilution. Such a setting implements a selection pressure for an autocatalytic system at the most basic level, as "death" by dilution can only be counteracted by a sufficient level of autocatalysis[8].

In these experiments, GUVs containing an initial full-length F1 starting concentration (1 or 2 μM) were mixed in equal amounts with separate vesicles containing either 10 μM Hyper-A (with 2.5 μM Cy5-A) or 15 μM B. To maximise content mixing, the tripartite vesicle population was then subjected to three consecutive freeze-thaw (FT) cycles (generation 0) followed by an incubation period of 1 h at 42 °C (generation 1). Subsequently, half of the resulting vesicles were

combined with an equal volume of fresh substrate vesicles and the entire process was repeated for several generations but without further addition of exogenous F1 (Fig. 3a). Feeding with fresh substrate vesicles was expected to provide the next generation of ribozymes with ample supply of substrates. After the initial generation 1, we observed about ~5% ligation of substrate A after the first cycle. These ligation yields remained close to constant in the following 7 generations suggesting that autocatalytic self-replication of F1 was able to persist at low levels despite ongoing serial dilution and content loss caused by repeated FT-cycling (Fig. 3b, Supplementary Fig. 19). To investigate if higher initial seed-concentrations might improve F1 self-replication, we increased the initial seed concentration of F1 to 2 μM and repeated the experiment. Here, the initial ligation yields for labelled F1 at generation 1 were ~10% and readily plateaued in subsequent generations at ~12%. From the fluorescence intensities of the source gels used in Fig. 2b, we can estimate that 3 consecutive FT-cycles lead to a content loss of ~45%. Taking the amount of fresh substrate added to the samples during each serial dilution step, the GUVs contain a total concentration of ~2.4 μM of fragment A (including Cy5-A) after each FT-induced content mixing step. Consequently, the self-replication of F1 results in a steady-state concentration of roughly 0.3 μM de novo synthesised F1 for reactions that were seeded with GUVs containing 2 μM F1. We could generally exclude that Hyper-A background activity was the main reason behind F1 formation: In control samples seeded with empty (0 μM F1) GUVs, background levels of de novo ligated F1 remained at ~1.8% (~0.04 μM) throughout the experiments (Fig. 3b). Thus, under the given experimental conditions, the initial seed concentration of F1 was the main determinant for the steady-state ligation levels observed across the seven generations. This behaviour suggests that the combination of ligation, dilution, and content leakage resulted in sub-saturating F1 concentrations for which ligation yields showed a near-linear dependency on the F1 starting concentration after 1 h incubation at 42 °C, similar to what we observed for sub-saturation F1 concentrations under bulk conditions (Supplementary Fig. 13c, d).

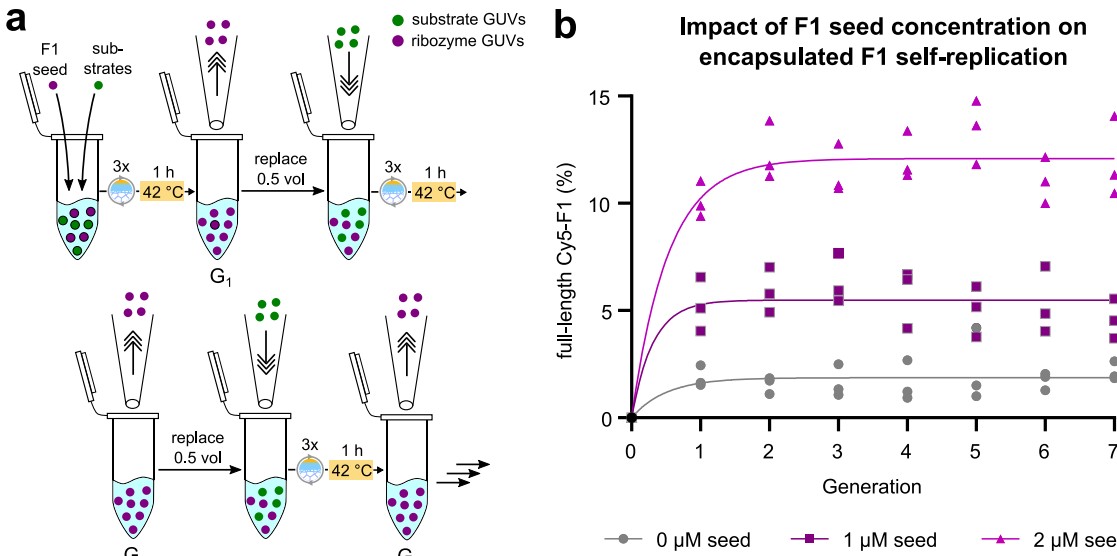

**Fig. 3 | Serial dilution of encapsulated self-replicating F1. a** Experimental setup of the serial dilution experiment based on the vesicle setup depicted in Fig. 2a. Briefly, after each cycle of amplification consisting of three successive freeze-thaw (FT) cycles followed by a 1-h incubation period at 42 °C, half of the volume was replaced with fresh substrate vesicles and the entire process was repeated. Only $G_0/G_1$ contained a specific seed concentration of unlabelled full-length F1. No exogenous full-length ribozyme was added in later generations. **b** Ligation yields of Cy5-labelled F1 after each generation. GUVs containing either 0 µM (grey circles), 1 µM (purple squares), or 2 µM (pink triangles) unlabelled F1 ribozyme as initial seed, 10 µM Hyper-A with 2.5 µM Cy5-A, or 15 µM B were combined in 1:1:1 ratio as shown in Fig. 2a. PAGE samples were taken before ($G_0$) and after ($G_1$) FT-induced content mixing and self-replication (42 °C for 1 h). In all further generations, PAGE samples were taken after 1:1 serial dilution of the previous generation with fresh substrate GUVs followed by three FT-cycles and the 1 h incubation at 42 °C as shown in **a**). Data were fitted with a single exponential for descriptive reasons. Every data point represents an independent experimental replicate.

In summary, our model system allowed us to generate an encapsulated RNA self-replicator capable of surviving multiple generations of serial dilution by spreading between vesicles through environmentally induced horizontal RNA transfer.

## Discussion

In this work, we explored how catalytic RNAs, including autocatalytic replicators, can propagate within a population of GUVs by exploiting transient, temperature-induced defects in the lipid bilayer. Following previous studies, we focused on periodic freeze-thaw (FT) cycles as the main drivers of content exchange that may have also occurred on the early Earth, e.g., in the wake of day-night cycles in cold regions. The high vesicle concentrations required to minimise content loss require a scenario that leads to continuous vesicle enrichment. A suitable environment in this context could have been shallow surface streams flowing through porous rocks. Here, the pores can serve as traps in which the vesicles accumulate before being exposed to freezing and thawing conditions. Studies have shown how microstructured posts can trap and accumulate GUVs[34], supporting the claim that porous rocks could potentially concentrate protocells. Moreover, a model Hadean microcompartment mimicking hydrothermal porous rock with a gas-water interface was shown to promote the encapsulation of prebiotically relevant biomolecules as well as aggregate vesicles[18]. The timescales and temperatures of plausible environmental temperature fluctuations clearly differ from the optimised lab conditions in our experiments (cooling at −80 °C to ensure homogeneous freezing and prevent super-cooled solutions followed by thawing at room temperature). Thus, future experiments may explore the mechanism established in this study under more plausible environmental temperature changes.

While transient content exchange between vesicles can aid the spread of autocatalytic replicators, it also opens the door for the propagation and takeover of the population by parasitic replicators[8]. Such parasites are short, fast replicating sequences without catalytic activity that are able to exploit existing replicases and therefore overwhelm autocatalytic replicators under bulk conditions by utilising all the available common precursor substrates. Several studies demonstrated that compartmentalisation provides a means to prevent parasitic takeover as physical boundaries can help to protect replicators from excessive dissemination of parasitic sequences throughout the protocell population[35,36]. Under additional selective pressure, both parts can coexist and even exhibit oscillatory dynamics[37]. Surprisingly, even transient compartmentalisation, similar to what occurs in our system, allows replicator survival in the presence of parasites if there is selection pressure for compartments containing active replicators[38]. The co-evolution of host and parasite RNA also allows for diversification of the replicator population aiding to prevent the collapse of the host replicating system[39]. Thus, it seems conceivable that replicating RNAs might have persisted in protocells despite their reliance on transient disruptions of their compartments for dissemination.

Testing this hypothesis will require a different replicator system, since the autocatalytic ligase used in this study is based on prefabricated RNA strands and therefore unable to evolve under the given conditions or provide a fitness advantage for the encapsulating vesicle. Such an RNA-vesicle co-selection might, for example, be based on mutual interactions between both biomolecules such that the paired system has a higher activity/stability than either component separately. RNA-lipid interactions, and in turn, lipid modulation of RNA catalytic activity, were shown to depend on several factors such as RNA sequence composition and hybridisation state, as well as membrane phase[40]. We have also noted a slight enhancement of ligation efficiency for the R3C ligase in the presence of vesicles (Supplementary Fig. 15). Compartmentalisation can also support the formation of crowded environments, which have long been known to enhance folding and catalysis of catalytic nucleic acids[41–44]. Furthermore, the encapsulation of functional RNA has been shown to preserve activity at high lipid concentrations that would otherwise inhibit unencapsulated RNA[45]. Although we do see signs and possible effects of potentially beneficial RNA-membrane interactions (Supplementary Figs. 8 and 15), further inquiries are necessary to elucidate the exact nature of modulation in

regards to the encapsulated autocatalytic replicator under the specific buffer and temperature conditions.

While our study was focussing on freeze-thaw cycling as the main driver of content exchange, different types of temperature fluctuations can have similar effects. For example, elevated temperatures have been shown to drive membrane phase transitions in fatty acid vesicles, enabling the generation of daughter protocells with reshuffled content[19]. Milder temperature fluctuations that did not necessarily involve phase-transitions in the bulk solvents might have also been of importance for early prebiotic processes. Whilst more complex to synthesise than fatty acid membranes, the superior stability of phospholipid membranes across pH, salt and temperature gradients makes them readily compatible with encapsulated nucleic acid catalysis[46–49]. Since content exchange relies on the membrane instability during the phase transition temperature of lipids, phospholipids with higher melting temperatures may also enable content exchange at temperatures above the freezing point of water.

Our findings support the argument that temperature fluctuations, including those involving eutectic ice formation, provide a suitable environment to drive a variety of reactions relevant to the emergence and evolution of life, potentially supporting a continuous path from the prebiotic chemical synthesis of monomers, their subsequent activation and non-enzymatic polymerisation to the point where they allow tertiary folds and associated catalysis including – but not limited to – autocatalytic replication. The simultaneous synthesis of nucleic acid precursors and phospholipids in an environment supporting the cyclical freezing and thawing of water could have resulted in compartment formation and encapsulation of molecules[50], while exchanging resources with the outer environment and other protocells by means of the temperature-induced membrane permeabilization.

## Methods

### In vitro transcription

ssDNA oligonucleotides were ordered from Integrated DNA Technologies (IDT). RNA was either ordered from IDT or transcribed in vitro (Supplementary Table 1). Briefly, a partially dsDNA template with double-stranded T7 promoter region was created by incubating equimolar amounts of the DNA oligomers at 85 °C and then placing them on ice. In vitro transcription (IVT) reactions (100 μL final volume) were carried out as follows: 1 μM DNA template, 50 mM Tris-HCl pH 7.8, 30 mM $MgCl_2$, 5 mM of each NTP (Jena Bioscience), 10 mM DTT, 2 mM spermidine (Sigma-Aldrich), 0.1 U Escherichia coli Inorganic Pyrophosphatase (NEB), and 0.5 μM of T7 polymerase (purified from a recombinant source in our laboratory). The reaction was incubated at 37 °C for 4–6 h. After column purification (Monarch RNA Clean-up Kit, NEB), the transcribed RNAs were gel purified after 12–20% denaturing PAGE. The transcript band was visualised by UV shadowing on a TLC plate, excised, crushed in a 2 mL tube with a syringe plunger and soaked overnight in 2 volumes of 0.3 M NaOAc pH 5.2 at 4 °C on a rotator. Gel debris was removed by spin filtration in 0.45 μm Spin-X cellulose acetate filters (Costar) for 2 min at 17,000 g, 4 °C. The eluted RNA was precipitated in 1.2 volumes of cold isopropanol, cooled 45–60 min at −20 °C and centrifuged for 90 min at 21,000 g, 4 °C. The supernatant was discarded and the pellet was washed with 1 mL of 80% ethanol and centrifuged another 20 min at 21,000 g, 4 °C. After discarding the supernatant, the pellet was dried under vacuum, suspended in ultrapure water and quantified on a Nanodrop measuring absorbance at 260 nm using their specific extinction coefficient calculated using OligoCalc[51] (http://biotools.nubic.northwestern.edu/OligoCalc.html). The correct length of all labelled and unlabelled RNA species in this paper was verified in a separated PAGE gel, which is shown in Supplementary Fig. 20.

### GUV components and general protocol

All GUVs were composed of 1-palmitoyl-2-oleoyl-phosphatidylcholine (POPC) as the phase transition temperature of this lipid (−2 °C) enables

reliable transient membrane permeabilization and content exchange during thawing episodes. Encapsulation was realised via an inverted emulsion transfer-based method. The emulsion transfer method relies on a density gradient between inner and outer phase achieved by equimolar (900 mM) solutions of internal sucrose and external glucose. The high sucrose concentration was shown to have only a negligible effect on substrate cleavage by HH-min (Supplementary Fig. 2) and on substrate ligation by the R3C ligase (Supplementary Fig. 10). In microscopy experiments, to distinguish between GUV populations, labelled lipids (Atto647N-DOPE) were added to the bilayers of the GUVs containing ribozyme whereas substrate GUV membranes remained unlabelled.

### Preparation of GUVs for microscopy

POPC lipids dissolved in chloroform were added to a glass vial containing chloroform to obtain 250 μL of a 6 mM solution. Fluorescent GUV membranes were obtained by adding 0.5 μL of a 1 mg/mL solution of Atto 647N-DOPE lipids in chloroform (~0.03 mol%). The chloroform was evaporated under nitrogen flow (10 min) and the lipids were dried under reduced pressure for 1 h to remove residual traces of chloroform. A lipid-mineral oil solution (400 μM lipids) was prepared by adding 3 mL of mineral oil to the dried lipids followed by sonication for 1 h at elevated temperature (40–60 °C). 750 μL of lipid-mineral oil solution was layered on top of 2 mL of outer phase solution in a 5 mL Eppendorf tube. Incubation for 15 min at RT allows lipid monolayer formation at the water-oil interface. Meanwhile, 750 μL of lipid-mineral oil solution and 15 μL of inner phase solution were combined in a 1.5 mL microcentrifuge tube. A water-in-oil emulsion was generated by rubbing the tube over a microtube rack. The water-in-oil emulsion was immediately and carefully pipetted to the biphasic mixture in the 5 mL Eppendorf tube. To generate lipid GUVs, gradual centrifugation (10 min at 300 g followed by 2.5 min at 1500 g) was performed to sediment the denser inverse micelles (surrounded by a lipid monolayer) through the interfacial lipid monolayer. Vesicle pellets at the tube bottom were visible by eye and could thus be easily taken up with a pipette. A 384-glass bottom microtiter plate was first passivated with 50 μL of a Pluronic F-127 solution (10 mg/mL) and then washed once with 50 μL of outer phase solution. Next, the pellet and outer phase solution (~ 50 μL) were transferred to the plate. Another GUV population was added and mixing of both populations was achieved by gently pipetting up and down (10 times). Before freeze-thawing, the plate was centrifuged for 5 min at 1500 g for GUV accumulation and high GUV densities. GUV density is inversely proportional to content loss, since GUVs that are not tightly packed together will lose most of their content to the outer phase. High GUV densities are thus required to minimise content loss and thus improve the overall dynamics of the system in question. If no measures of GUV accumulation are undertaken, content loss can increase up to 80%. All steps, including mineral oil were performed in a custom-made glove box under reduced humidity (<10% relative humidity) except for the centrifugation step for GUV formation.

### Freezing and thawing of microtiter plate samples for microscopy

Freezing and thawing were performed at high cooling and low heating rates for all experiments. These parameters were optimised in our previous article[29]. Briefly, the fast-cooling minimises vesicle rupture during freezing, whereas the slow heating provides more time for content exchange during a thawing episode. The result is a high exchange efficiency and a low content loss. Before sample freezing, glass bottom microtiter plates were sealed with the aluminium-coated sealing film ROTALIBO®. A block made of stainless steel pre-cooled in a −80 °C freezer was pressed against the glass bottom leading to sample freezing within a few seconds. Plates were then stored at −80 °C for up to 30 min, taken out and thawed at RT while holding them at an angle

to avoid GUV dispersal, thus maintaining high GUV densities required for an efficient content exchange. In time-lapse experiments (Supplementary Fig. 14b–d, Supplementary Movies 1–3), plates remained attached to the microscope stage during freeze-thawing. Here, fast-freezing was achieved by liquid nitrogen poured into the wells surrounding the sample well. Condensation on the glass bottom was avoided by blowing in $N_2$ gas from below to enable sample access at sub-zero temperatures with a dry objective. Sample thawing at RT started once the liquid nitrogen was fully evaporated.

### Preparation of GUVs for PAGE assays with freezing and thawing
POPC lipids dissolved in chloroform were dried under nitrogen flow, and placed under vacuum for 1 h to remove traces of chloroform. The dried lipids were subsequently dissolved in light mineral oil (Carl Roth) to obtain a final concentration of 400 µM, incubated at 60 °C for 5 min then vigorously vortexed for 1 min, and finally sonicated for 1 h at 40–60 °C in a water bath to disperse aggregates and completely resuspend dried lipids. Next, 1.5 mL tubes were prepared with 600 µL of outer phase containing ribozyme buffer (as specified per experiment) and 900 mM glucose, layered with 300 µL of lipids in oil suspension and was allowed to rest for 30 min to properly form the lipid monolayer at the oil-water interface. The emulsion was prepared by adding 24 µL inner phase solution (containing 900 mM sucrose) to 400 µL of lipids in oil solution and mechanically agitating the tube over a tube rack until the suspension turned milky white. The emulsion was carefully transferred to the oil layered on the outer phase, and spun at 300 g for 5 min. The vesicle pellet was recovered by piercing the bottom of the open tube with a needle and ejecting it into a clean tube by closing the cap. GUV populations were united, ~700 µL fresh glucose outer phase was added and the vesicles were spun at 2000 g for 5 min. Most of the supernatant was discarded, the pellet was resuspended in ~15 µL buffered glucose outer phase, and 10 µL samples were placed in PCR tubes and spun again at 2000 g for 5 min to form a vesicle pellet. Control samples were held at room temperature (RT) and the rest were subjected to cycles of freezing and thawing as follows: 5 min in aluminium block at −80 °C, 5 min at room temperature, and incubation at the desired reaction temperature. When mentioned, the freeze-thaw cycle was applied multiple times.

### RNA recovery from GUVs and PAGE analysis
To recover the encapsulated RNA from GUVs and analyse activity via polyacrylamide gel electrophoresis (PAGE), the following protocol was employed: samples of 5–10 µL were added to 500 µL of unbuffered glucose outer phase, mixed by inverting the tube and spun at 2000 g for 5 min. 450 µL of supernatant was then replaced with an equal volume of fresh unbuffered glucose outer phase. The wash steps serve to exchange the outer phase and remove any biomolecules or salts that leaked from GUVs. After exchanging the outer phase, the GUVs were pelleted by spinning at 2000 g for 5 min, the supernatant was discarded leaving only around 10 µL of pellet, and 10–20 µL of GUV loading buffer was added (containing 25 mM EDTA, 0.3% Triton X-100, 0.01% bromophenol blue, 95% formamide) and vortexed vigorously for 5–10 s. Finally, the samples were denatured by incubating for 5 min at 85 °C and placed on ice before 12–20% denaturing urea-PAGE analysis. Gels were imaged immediately after electrophoresis on an Azure RGB Sapphire and analysed using Azure Spot (2.2.167).

### Hammerhead assay in GUVs for microscopy
Two GUV populations encapsulating the hammerhead buffer (900 mM sucrose, 20 mM Tris-HCl pH 8.3, 4 mM MgCl₂) were prepared and combined in a 1:1 ratio as described above. One population contained the substrate (HH-FQ-Sub−2.5 µM) and the other membrane-labelled population contained either hammerhead ribozyme (HH-min−5 µM), the inactive ribozyme variant (HH-mut−5 µM) or no ribozyme (negative control−Neg Ctrl). The outer phase consisted of hammerhead buffer and 900 mM of glucose to ensure both a density gradient and isosmotic conditions between inner and outer phase. Sample freezing and thawing were applied as described above. Images were taken at room temperature before and after freeze-thawing.

### Hammerhead assay in GUVs with PAGE analysis
GUVs encapsulating hammerhead ribozymes were prepared as described above using hammerhead buffer (900 mM sucrose, 20 mM Tris-HCl pH 8.3, 4 mM MgCl₂) and 5 µM of either HH-min. Substrate GUVs were also prepared the same way except that it included 2.5 µM Cy5-tagged substrate RNA instead of ribozyme. The outer phase was the hammerhead buffer but with 900 mM of glucose rather than sucrose, which is necessary for the density gradient. The two GUV populations were united and washed once with 700 µL of buffered outer phase followed by spinning at 2000 g for 5 min. The volume was reduced to ~25 µL and split into two samples of 10 µL each. One sample was a control held at room temperature while the other was subjected to a cycle of freezing and thawing. Both samples were subsequently incubated at 37 °C for 1 h before the GUVs were washed, the RNA was recovered from the GUVs and analysed by 20% denaturing PAGE, as described above.

### Generation of inactive R3C reporter substrate
During an initial characterisation of the F1 self-replicator under bulk conditions, we observed substantial background formation of F1 from A and B even in absence of the full-length ribozyme suggesting a residual activity of both fragments, likely resulting from their association even in the absence of F1 ribozyme (Supplementary Figs. 10 and 11). However, we observed no background ligation of the full-length ribozyme in assays containing a variant of substrate A that was derived from the parental wildtype ribozyme E1 and which differs only by the single transition G38A from the F1-substrate A (Supplementary Fig. 1c, red box). We used the inactive, Cy5-labelled E1 derived substrate as a reporter (referred to as Cy5-A) and the F1-derived substrate as the active substrate that contributes to autocatalysis (referred to as Hyper-A).

### FT-dependent activity of R3C ligase
The different GUV populations were prepared independently as described above. Briefly, ribozyme GUVs encapsulated 1 or 2 µM of ribozyme F1 in R3C buffer (50 mM EPPS pH 8.5, 20 mM MgCl₂, 900 mM sucrose). Substrate A GUVs contained a mixture of 10 µM Hyper-A and 2.5 µM Cy5-A in R3C buffer and substrate B GUVs encapsulated 15 µM of B in R3C buffer. Empty GUVs were used as a negative control and contained only R3C buffer with no RNA. The outer phase also contained the R3C buffer but with 900 mM glucose instead of sucrose. The GUV populations were combined in a 1:1:1 ratio to obtain a starting sample with both substrate vesicles and either ribozyme GUVs or empty GUVs as the control. The GUVs were washed once with ~700 µL buffered glucose outer phase and centrifuged at 2000 g for 5 min. The supernatant was discarded, and the pellet was resuspended in ~15 µL buffered glucose outer phase and 10 µL were transferred to PCR tubes and subjected to different numbers of FT cycles (none, one, two or three successive cycles) followed by a 1-h incubation at 42 °C. Samples were subsequently washed twice in 500 µL unbuffered glucose and prepared for 12% denaturing urea PAGE as described above.

### Serial dilution of autocatalytic R3C ligase encapsulated in GUVs
The GUV populations were prepared as described above with the same RNA concentrations and buffer conditions. A population of substrate vesicles (separate GUV populations containing either 10 µM Hyper-A and 2.5 µM Cy5-A or 15 µM of B in R3C buffer) was prepared to be used as a fresh feedstock. The reaction was started by combining 10 µL of GUVs containing unlabelled F1 with 20 µL containing equal amounts of both substrate GUVs (initial mixing ratio 1:1:1), followed by a wash step

and centrifugation at 2000 g for 5 min. After discarding 15 μL of the supernatant, settled GUVs were resuspended, and a 5 μL sample was taken and washed in 500 μL of 900 mM unbuffered glucose and subsequently resuspended in 20 μL of the GUV loading buffer ($G_0$ PAGE sample). A second 5 μL sample for PAGE analysis was taken from the remaining 10 μL GUV mixture after three consecutive FT-cycles followed by incubation at 42 °C ($G_1$). In all further generations, 5 μL of the GUVs sample from the previous generation was mixed 1:1 with 5 μL of feedstock substrate GUV containing equal amounts of substrate A and B (see above). These new generations were centrifuged at 2000 g for 5 min and subjected to 3 FT cycles followed by incubation at 42 °C after which 5 μL were removed for PAGE analysis. After all generations were sampled, they were heat denatured and loaded on a 12% denaturing urea-PAGE and analysed as above. Data points were fitted to a single exponential using GraphPad Prism for descriptive purposes.

### Laser scanning confocal fluorescence microscopy

All images were taken on a laser scanning microscope LSM 780 (Carl Zeiss, Germany) equipped with a water immersion objective (C-Apochromat 40×/1.20 W) for acquisition at room temperature before and after FT, or a dry objective (Plan-Apochromat 40×/0.95) for time-lapse imaging during FT-cycling. Samples were excited at 488 nm for FAM/Atto-488, 561 nm for TAMRA/Alexa Fluor-568 excitation and 633 nm for Atto-647N. At room temperature, colours were excited separately to avoid cross-talk whilst time-lapse experiments required simultaneous illumination. Images were typically recorded using a 1 airy unit pinhole, a resolution of 512 × 512 pixels and pixel dwell times ranging from 12–25 μs. Time-lapse imaging was typically performed with 3 μs pixel dwell times and 1 s intervals. For tile-scan imaging, a group of individual adjacent image fields (tiles) was recorded and stitched together by the Zeiss software (Zen Black, v13.0.2.518).

### Image analysis

All confocal images shown were analysed using Fiji (v1.53c). For visual presentation, images were merged from separately recorded channels and adjusted for brightness and contrast. Brightness and contrast adjustments were applied homogenously for all acquired images. Image stacks derived from time-lapse experiments were analysed by defining regions of interest using the built-in ROI manager. Data processing and plotting were performed using OriginPro (2020b, v9.7.5.184).

### Reporting summary

Further information on research design is available in the Nature Portfolio Reporting Summary linked to this article.

## Data availability

The data that support the findings of this study are available within the main text and its Supplementary Information file. Source data are provided with this paper. The source data for all charts and graphs are provided in an Excel document where each worksheet contains the relevant data for each figure and supplementary figure. In addition, all uncropped gel scans used for the figures are provided on the respective worksheet in the Excel document. Data is also available from the corresponding author upon request. Source data are provided with this paper.

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

## Acknowledgements
Funded by the Deutsche Forschungsgemeinschaft (DFG, German Research Foundation) – Project-ID 364653263 – TRR 235, Project P14 (H.M and P.S.). H.M. is grateful for support from the European Research Council (ERC) under the European Union's Horizon 2020 research and innovation programme (grant agreement no. 802000, RiboLife).

## Author contributions
H.M. and P.S. conceived the project. E.S and B.P. designed experiments, collected the data, and performed the analysis. All authors wrote the paper. All authors read and approved the final manuscript for publication.

## Funding

## Competing interests
The authors declare no competing interests.
