## [Peer Review File · Nature Communications]

Periodic temperature changes drive the proliferation of self-replicating RNAs in vesicle populationsREVIEWER COMMENTS

Reviewer #1 (Remarks to the Author):

Review of manuscript titled "Exchange, catalysis and amplification of encapsulated RNA driven by periodic temperature changes" by Salibi et. al.

Salibi et. al. have used their previously characterized high-throughput system of GUV-based protocells, for demonstrating how geologically relevant freeze-thaw (FT) cycles, results in the assembly of active HH ribozyme from inactive precursors encapsulated in disparate populations of lipid vesicle. Pertinently, the exchange of contents result in an encapsulated ribozyme that potentially overcomes the problem of 'freezing-induced content loss', and also dilution in scenarios where the system is subject to a dilution protocol, simulating what might readily advent on the early Earth. Further, in the R3C-based experiments, there is a potential uncoupling of the membranes' growth and division process, from that of the replication of the nucleic acid component. This they argue is facilitated by avoiding "death" by dilution', which is made possible by repeatedly infusing GUVs with a certain concentration of substrates to the reaction system. This showed that there is a maintenance of a basal level of autocatalysis, sustaining the catalytic system despite the dilution-based selection pressure applied on the enzyme. Overall, the scientific work under consideration is sound with the experimental evidence reported being adequate, and the authors producing a really well written article that was a pleasure to read. Eutectic phases have long been explored in the origins of life field, especially in the context of the test system that the authors have used to study their hypothesis. The exciting aspect presented in here is the compartmentalization angle, which makes the system under experimentation a sustainable one for catalysis and amplification. Nevertheless, it would have been nice if the authors also included an experiment that explicitly quantified the benefit of such compartmentalized systems in a eutectic phase (vs the bulk phase). Overall, I assess the quality of this manuscript as being suitable for publication. Nonetheless, below are comments that the authors should address to improve the overall comprehensiveness of the manuscript.

Specific comments:

1. The authors detail few studies done in eutectic phases/ice-water matrices in the Discussions section between lines 295-301. This part could potentially be moved to the introduction section as it provides a clear context for the basis of their work.
2. There is an observable decrease in the red signal that is coming from the membrane as the freezing progresses. Is this due to loss of membrane or the rupture of vesicles? This further gives rise to another pertinent question: Is the reaction happening in the absence of GUV and then getting re-encapsulated during thawing? Since, it is a significant contributor to the main results, it would be important for the authors to discuss this explicitly in the manuscript.
3. Page 6, lines 144-147: The invocation of non-homogeneous content exchange should, in principle, also apply to the hammerhead ribozyme assay even if it was somewhat less substantial. The authors should share their observations in this regard.
4. In the results described in Fig. 4, it appears the seeding was done in each generation. It will be good to mention this explicitly in lines 242-245 as it has significant impact on the experimental outcome.
5. In Fig. 4, the 2 μ M seeded reaction results shown in panels b and c, have higher error bars on an average, especially in c. What is possibly leading to this? Will be good for the authors to discuss this briefly.
6. In lines 387-389, the authors mention that they followed a specific protocol to maintain high GUV densities. Nonetheless, if one comes at this from a very prebiotic perspective, the accumulation and high density of GUVs might have not been probable in most scenarios. It would, therefore, be good to mention how much of difference is there in the exchange reaction, if this physical parameter is not

accounted for.

7. On a similar note, in the methods section the authors mention that the cooling step is rapid while the thawing is gradual (lines 391-395) – is there a rationale behind such a method? On the early Earth, it is reasonable to expect that both of these processes would have been comparable timescale wise. Given that the rate of cooling and heating can directly affect the dynamics of the reaction, this is a point that should be clarified.

8. In the supplementary Fig. 14, the authors have monitored effect of FT on GUV. It will be nice to include a movie in the SI for this experiment as it is a very impressive feat to monitor the whole process.

Reviewer #2 (Remarks to the Author):

Summary

The manuscript describes an experimental model system to address an important question regarding the earliest ribozyme-based cells within protocells: How could such systems have grown and duplicated? The study uses freeze-thaw cycles can lead to the mixing of content from separate lipid vesicles, and that this phenomenon can be used to 'feed' the replicating RNAs within the vesicle to allow the replication of the RNAs. The study uses two ribozymes as model systems, a hammerhead ribozyme and a ligase ribozyme; the latter is used in the format of ligating two substrate RNAs as well as in an autocatalytic format displaying self-replication from two RNA fragments.

This manuscript is an exciting advance towards generating an 'RNA world' in the lab, and I support its publication in Nature Communications if three points are addressed: 1 - what are the benefits and cost of the vesicle system to the self-replicating ribozyme system, 2 - focus the manuscript on the exciting part of the study, and 3 - tone down claims of prebiotic plausibility regarding the -80°C freeze treatments.

Major points

- The manuscript needs to address, in the discussion, the point that, if vesicles can exchange genetic material then the vesicles will also not protect against molecular parasites. What are the implications for this mechanism? Similarly, the manuscript needs to address the point what benefits would the lipid membrane provide to the autocatalytic ribozyme system. For example, what is the self replication rate if the vesicles are left out (i.e. what is the vesicle-mediated decrease in activity) and how much of the presumable vesicle benefit.

- The manuscript takes a while to come to the exciting part, which is the demonstration of R3C-based replicator autocatalysis within the vesicles. This means that the manuscript may benefit if (1) the study of hammerhead ribozymes would be omitted, (2) the study of ligase ribozymes mediating the ligation of two short substrates would be omitted, and (3) most of the discussion of Hyper-A and wildtype ribozyme E1 (lines 179-194) would be moved to the materials & methods section; here, one or two sentences on the used ribozyme, substrate, and concentrations are sufficient (and moving what was not used).

- In a number of cases, the used experimental conditions are implied to be prebiotically plausible - with freeze-thaw cycles at -80°C. The claims of prebiotic plausibility need to be clarified and/or toned down. lines 59 and 94: The statement that 'less extreme freezing conditions' were used in the study compared to the liquid nitrogen freezing conditions in reference 22 is misleading when no additional detail is added - because the condition of -80°C stated in line 94 is not very prebiotically plausible either. Please add the detail of -80°C in line 59 or otherwise make the statement in line 59 more transparent. Similarly, the terms 'plausible geochemical environment' and 'geological scenario' in the

discussion (lines 288 and 290) don't describe the used conditions.

Minor points

Line 39: The term 'readily available' may be an overstatement because the synthesis is not very efficient, and needs to proceed in a very different environment than the synthesis of all other compounds. I suggest replacing with 'prebiotically plausible' or similar.

line 85: Please define 'HH-FQ-sub' the first time this term is used.

Lines 95 and following: Could a statement be made what fraction of cleaved Hh ribozyme substrate would correspond to the observed increase in fluorescence? While the PAGE analysis addresses that question it would be nice to have an estimate what fraction of substrate cleavage would be consistent with the observed increase in fluorescence.

Figure 1: Please replace the acronym 'NC' with a more intuitive acronym (all other acronyms such as HH-mut and HH-min are intuitive). This will make it easier for the reader to understand the figure.

I suggest reminding the reader every now and then that "FT" means 'freeze-thaw'. The reader is not using this acronym on a daily basis and therefore needs more time to understand the sentences (it took me three readings of the sentence and one reference to line xx in order to understand line 195-197).

REVIEWER COMMENTS

Reviewer #1 (Remarks to the Author):

Review of manuscript titled “Exchange, catalysis and amplification of encapsulated RNA driven by periodic temperature changes” by Salibi et. al.

Salibi et. al. have used their previously characterized high-throughput system of GUV-based protocells, for demonstrating how geologically relevant freeze-thaw (FT) cycles, results in the assembly of active HH ribozyme from inactive precursors encapsulated in disparate populations of lipid vesicle. Pertinently, the exchange of contents result in an encapsulated ribozyme that potentially overcomes the problem of ‘freezing-induced content loss’, and also dilution in scenarios where the system is subject to a dilution protocol, simulating what might readily advent on the early Earth. Further, in the R3C-based experiments, there is a potential uncoupling of the membranes’ growth and division process, from that of the replication of the nucleic acid component. This they argue is facilitated by avoiding “‘death” by dilution’, which is made possible by repeatedly infusing GUVs with a certain concentration of substrates to the reaction system. This showed that there is a maintenance of a basal level of autocatalysis, sustaining the catalytic system despite the dilution-based selection pressure applied on the enzyme. Overall, the scientific work under consideration is sound with the experimental evidence reported being adequate, and the authors producing a really well written article that was a pleasure to read. Eutectic phases have long been explored in the origins of life field, especially in the context of the test system that the authors have used to study their hypothesis. The exciting aspect presented in here is the compartmentalization angle, which makes the system under experimentation a sustainable one for catalysis and amplification. Nevertheless, it would have been nice if the authors also included an experiment that explicitly quantified the benefit of such compartmentalized systems in a eutectic phase (vs the bulk phase). Overall, I assess the quality of this manuscript as being suitable for publication. Nonetheless, below are comments that the authors should address to improve the overall comprehensiveness of the manuscript.

We thank the reviewer for the positive evaluation, kind comments and time invested into reading the manuscript. Below and in the revised manuscript, we have addressed the comments and concerns raised. Briefly, we abridged the results section and elaborated more in the discussion. Moreover, three new SI figures were added that address the fraction of cleaved substrate in microscopy studies and the effect of vesicles on RNA catalysis as well as the timing of the reaction.

Specific comments:

1.1. The authors detail few studies done in eutectic phases/ice-water matrices in the Discussions section between lines 295-301. This part could potentially be moved to the introduction section as it provides a clear context for the basis of their work.

We thank the reviewer for this helpful comment. The studies about eutectic ice phases have been moved from the discussion (previously lines 292 – 302) to the introduction (lines 53 – 63). The

beginning of the next sentence was remodelled such that it connects better with the previously inserted section detailing the studies done in eutectic ice.

1.2. There is an observable decrease in the red signal that is coming from the membrane as the freezing progresses. Is this due to loss of membrane or the rupture of vesicles?

We thank the referee for addressing these observations. The fluorescence decrease of membrane lipids likely results indeed from the loss of membrane components / rupturing during freezing and thawing (as shown in Supplementary Figure 17). We have now also added supplementary movies that illustrate the fate of individual GUVs during freezing / thawing that experience effects such as fragmentation, protrusion or osmotic stress depending on their local environment (Supplementary Movies 1-3).

1.3 This further gives rise to another pertinent question: Is the reaction happening in the absence of GUV and then getting re-encapsulated during thawing? Since, it is a significant contributor to the main results, it would be important for the authors to discuss this explicitly in the manuscript.

We thank the reviewer for raising this point. From control experiments, we know that the amount of ligation that happens during thawing is very low compared to the post-thawing incubation at 42°C. This negligible amount of ligation during the frozen conditions or during thawing is even observed for high bulk concentrations that resemble the intravesicular concentrations (now shown in the new Supplementary Figure 18).

Taken together, we conclude that the vast majority of catalysis happens inside GUVs and not in the absence of GUVs before thawing or during early thawing. Furthermore, for RNA that is not re-encapsulated, it seems unlikely that it will react and get encapsulated during later FT-cycles due to dilution effects.

We have now included this important observation in the results section (p 7, lines 188-191):

“The freeze-thaw cycles also result in a temporal decoupling of RNA propagation and actual self-replication: since R3C ligase is nearly inactive under both frozen conditions and during thawing (Supplementary Figure 18), actual self-replication occurs predominantly under encapsulated conditions.”

3. Page 6, lines 144-147: The invocation of non-homogeneous content exchange should, in principle, also apply to the hammerhead ribozyme assay even if it was somewhat less substantial. The authors should share their observations in this regard.

We thank the reviewer for pointing this out. The inhomogeneous exchange of content is indeed also observed for the hammerhead ribozyme system, although to a much lesser extent. Based on observations from a mechanistic study (to be published elsewhere), we believe that the degree of content homogeneity after exchange depends strongly on the local microenvironment of frozen GUVs: during both freezing and thawing, there is only a limited time available for diffusion processes due to ice formation as well as the transient nature of membrane defects during thawing. These short time windows limit the exchange of contents to local clusters of GUVs. Thus, effects that influence the local GUV clusters such as size, composition or vesicle rupture may result in GUVs with a different RNA-composition compared to GUVs from other clusters. This effect is less drastic in

binary GUV population (such as for the hammerhead system) than in system where catalysis depends on three RNA components (such as for the ligase system). The differences in the local microenvironments are also exemplified in the new Supplementary Movies 1-3.

To clarify this idea, we omitted from the text ‘Unlike the assays with the hammerhead system...’ (previously lines 144-145), and added an explanatory section about the emergence of non-homogenous content exchange (lines 145 -155): ‘We noticed substantial heterogeneity ...’

4. In the results described in Fig. 4, it appears the seeding was done in each generation. It will be good to mention this explicitly in lines 242-245 as it has significant impact on the experimental outcome.

We apologise if our description of the protocol gave a wrong impression regarding the differences between “seeding” and “feeding”. In fact, we do not seed the reaction with new F1 ribozyme after each generation but only before the first FT-cycle. In the following generations, the system gets diluted 1:1 with fresh vesicles containing either substrate A or substrates B (“feeding”).

To highlight this, we have added the following text to the manuscript (p.9 lines 217-221): “To maximise content mixing, the tripartite vesicle population was then subjected to three consecutive freeze-thaw (FT) cycles (generation 0) followed by an incubation period of 1 hour at 42 °C (generation 1). Subsequently, half of the resulting vesicles were combined with an equal volume of fresh substrate vesicles and the entire process was repeated for several generations but without further addition of exogenous F1 (Figure 3a).”.

5. In Fig. 4, the 2µM seeded reaction results shown in panels b and c, have higher error bars on an average, especially in c. What is possibly leading to this? Will be good for the authors to discuss this briefly.

We thank the reviewer for this careful observation. The higher error bars for the 2 µM seeded reaction were likely due to the complications of handling the GUVs, the steps required to recover the intravesicular RNA content, and fluctuations during the PAGE experiments/analyses, as sample volumes and therefore intensities of the lanes can be very low.

To improve the data quality and reduce the effect of noise, we decided to repeat the serial transfer experiment with an improved protocol (e.g. more RNA substrate including Cy5-A reporter substrate, increased overall RNA concentration for PAGE experiments, seven generations, improved recovery) to obtain higher quality data. We also decided to show the direct cumulative yields of the labelled ligase (Cy5-F1) rather than using the estimated intravesicular ligase concentration as this gives a more direct readout of the de novo replicated ligase throughout the generations and does not include estimations based on the concentration of the unlabelled seed ligase. In our previous analysis we also did not report the estimated ligase concentration normalised to the whole GUV sample volume. In our new brief estimate of the steady-state ligase concentration, we assumed a nearly homogeneous distribution of RNA across all vesicles after the three FT cycles that were carried out after each feeding step from generations 2 - 7. The new data is described in detail on p 9, lines 223-245.

The results from the new experiments (each from three independent replicates) are now presented in Figure 2b and 3b (as well as the accompanying gel images in the supplementary information) and

show that stable yields of labelled ligase are reached after 2-3 FT-cycles and that these yields are maintained by the system throughout further generations of serial dilutions. Moreover, as in the previous experiments, the results emphasize that the initial seed amount of ribozyme determines the steady state of replicating ribozymes in all further generations. Compared to the previous data, the improved protocol now also enables “survival” of replicators above background levels for reactions that were seeded with GUVs containing just 1 μM initial F1. A detailed description and analysis were added to the main text.

6. In lines 387-389, the authors mention that they followed a specific protocol to maintain high GUV densities. Nonetheless, if one comes at this from a very prebiotic perspective, the accumulation and high density of GUVs might have not been probable in most scenarios. It would, therefore, be good to mention how much of difference is there in the exchange reaction, if this physical parameter is not accounted for.

We thank the referee for this bringing this to our attention. By working at high density of GUVs, we were able to reduce content loss to the outer phase to $\sim 20\%$. At low GUV densities, content loss can increase up to 80% per FT-cycle, making it of course unlikely that a system will exhibit sustained replication over multiple generations. More mechanistic data that investigates the effects of parameters such as vesicle density on GUV content exchange / loss is in preparation and will be submitted in a separate article soon, which is why the topic is only briefly addressed in this work.

From a prebiotic perspective, different scenarios could provide the required accumulation of vesicles over time such as trapping in a geological microenvironment that resemble artificial microfluidic systems (e.g. Yandrapalli and Robinson, <https://doi.org/10.1039/C8LC01275J>) or thermogravitational accumulation whose potential for prebiotic chemistry has been demonstrated now for a variety of different systems (Morasch et al., <https://doi.org/10.1038/s41557-019-0299-5>).

We have also added a more thorough discussion of the prebiotic scenarios that might be required to achieve high vesicle densities in an actual environment (see response to comment 2.4 from referee 2), p.10 lines 266-270: “The high vesicle concentrations required to minimize content loss require a scenario that leads to continuous vesicle enrichment. A suitable environment in this context could have been shallow surface streams flowing through porous rocks. Here, the pores can serve as traps in which the vesicles accumulate before being exposed to freezing and thawing conditions”

7. On a similar note, in the methods section the authors mention that the cooling step is rapid while the thawing is gradual (lines 391-395) – is there a rationale behind such a method? On the early Earth, it is reasonable to expect that both of these processes would have been comparable timescale wise. Given that the rate of cooling and heating can directly affect the dynamics of the reaction, this is a point that should be clarified.

The heating and cooling rates were chosen based on the previous referenced paper (Litschel et al. 2018, [29]) that investigated these effects on content exchange of short fluorescently labelled DNA strands. The main reason for incubation in a metal block precooled to $-80\text{ }^{\circ}\text{C}$ is to ensure homogeneous freezing of the solutions. Small volumes in the μL range sometimes remain supercooled if they are incubated at higher temperatures (i.e. $-20\text{ }^{\circ}\text{C}$) due to the lower probability of ice nucleation events. We have not yet attempted to control the freezing rate by a more elaborate setup. Of course, it cannot be ruled out, that depending on the cooling rate, the rupture of the

membranes during freezing might lead to further unfavourable content loss of the GUVs. We are aware of that limitation of our work.

We have elaborated on these points in the material and methods section (lines 390-393): 'These parameters were optimised in our previous article [29]. Briefly, the fast cooling minimises vesicle rupture during freezing, whereas the slow heating provides more time for content exchange during a thawing episode. The result is a high exchange efficiency and a low content loss.'

We would also like to emphasize that in our serial transfer experiments each generation was exposed to three FT cycles each. As a result, the cumulative content loss was approximately 45-50% per generation. Nevertheless, the replicators were able to maintain their steady-state concentration despite content loss AND serial dilution, suggesting that exponentially amplifying RNA replicators are able to "survive" in GUV populations even when exposed to rather harsh environmental factors that continuously reduce their concentration.

8. In the supplementary Fig. 14, the authors have monitored effect of FT on GUV. It will be nice to include a movie in the SI for this experiment as it is a very impressive feat to monitor the whole process.

We thank the reviewer for this suggestion and have included representative movies in the SI data, as Supplementary Movies 1-3 for the respective time courses shown in Supplementary Figure 17 (previously Supplementary Figure 14d). The movie (new Supplementary Movie 3) associated with the images shown in Supplementary Figure 17d initially contained a number of low-quality images/frames between the timeframes 117 s and 153 s due to focussing issues. For this reason, the movie starts at 153 s and neither includes the first image of the shown image series ('117 s') nor the moment of sample freezing.

Reviewer #2 (Remarks to the Author):

Summary

The manuscript describes an experimental model system to address an important question regarding the earliest ribozyme-based cells within protocells: How could such systems have grown and duplicated? The study uses freeze-thaw cycles can lead to the mixing of content from separate lipid vesicles, and that this phenomenon can be used to 'feed' the replicating RNAs within the vesicle to allow the replication of the RNAs. The study uses two ribozymes as model systems, a hammerhead ribozyme and a ligase ribozyme; the latter is used in the format of ligating two substrate RNAs as well as in an autocatalytic format displaying self-replication from two RNA fragments. This manuscript is an exciting advance towards generating an 'RNA world' in the lab, and I support its publication in Nature Communications if three points are addressed: 1 - what are the benefits and cost of the vesicle system to the self-replicating ribozyme system, 2 - focus the manuscript on the exciting part of the study, and 3 - tone down claims of prebiotic plausibility regarding the -80°C freeze treatments.

We thank the reviewer for the support and critical reading of our manuscript. The points raised by the reviewer have been addressed below and in the revised manuscript. Briefly, we have abridged the results section and moved a main figure (previously Figure 2) to the SI (now Supplementary Figure 7). Moreover, we have followed the referee's suggestions (see below) and tackled how vesicles and encapsulation affect the autocatalytic ribozyme system, toned down the claims of prebiotic plausibility and clarified the point of using a non-prebiotically plausible temperature (-80 °C) to perform the experiments.

Major points

2.1 The manuscript needs to address, in the discussion, the point that, if vesicles can exchange genetic material then the vesicles will also not protect against molecular parasites. What are the implications for this mechanism?

We thank the reviewer for bringing this point to our attention. Indeed, exchange between vesicle populations will also allow potential parasitic sequences to spread across the population. As our ribozyme system cannot evolve parasites in its current form and is therefore not vulnerable towards parasitic take over, we are currently not in the position to address this important question. However, it was shown before by Matsumura et al. 2016 (<https://doi.org/10.1126/science.aag1582>), that even transient compartmentalization (as it occurs in our system) can prevent parasitic take over in evolvable systems as long as there is selection for functional replicators. Currently, there are no autocatalytic ribozymes systems available that can also evolve similar to a protein-based system (such as Qbeta replicase as shown by various works from the Ichihashi lab) as they are all based on pre-fabricated RNA elements. Furthermore, our system in its current form has no specific selection pressure for functional replicators. This could, for example, be implemented by a coupling of R3C replication to vesicle properties that would be beneficial for the GUV to "survival" under selection pressure. However, these kind of population dynamics are not the focus of our current work but might be tackled in future projects. We have included a new paragraph in the Discussion section that illuminates the parasite problem and potential solutions (p.11 lines 276 – 290): "While transient content exchange between vesicles can..."

2.2 Similarly, the manuscript needs to address the point what benefits would the lipid membrane provide to the autocatalytic ribozyme system. For example, what is the self-replication rate if the vesicles are left out (i.e. what is the vesicle-mediated decrease in activity) and how much of the presumable vesicle benefit.

To address the interactions of lipid membranes and RNA catalysts, we added a paragraph in the discussion that explicitly deals with this topic (p.11 lines 291 – 307): ‘Testing this hypothesis will require a different replicator system, since ...’

Moreover, we have included a new SI figure, specifically Supplementary Figure 15, where the activity of the R3C ligase was tested in the absence and presence of low GUV densities (0 – 50% v/v of the total reaction). Noticeably, in presence of a low concentration of GUVs, the rate of ligation is slightly increased, suggesting that these lipid concentrations and temperature favour the formation of product. As mentioned in the newly added text, this effect was shown to be reversed at high lipid concentrations, presumably because the RNA-lipid interactions become increasingly inhibitory.

2.3 The manuscript takes a while to come to the exciting part, which is the demonstration of R3C-based replicator autocatalysis within the vesicles. This means that the manuscript may benefit if (1) the study of hammerhead ribozymes would be omitted, (2) the study of ligase ribozymes mediating the ligation of two short substrates would be omitted, and (3) most of the discussion of Hyper-A and wildtype ribozyme E1 (lines 179-194) would be moved to the materials & methods section; here, one or two sentences on the used ribozyme, substrate, and concentrations are sufficient (and moving what was not used).

We thank the reviewer for the recommendations. We have omitted the second part of the results that contained the microscopic study of the fluorogenic ligase system. To avoid shortening the manuscript excessively, we condensed the sections describing the data obtained with the hammerhead ribozyme and the description of the autocatalytic self-replicator, while moving the more experimental details to the materials and methods section. However, we chose to retain but abridge the hammerhead study because it was fundamental for us as a proof-of-concept experiment and to continue investigating the system. More specifically, the following changes have been implemented:

- 1. A part of the introductory hammerhead section was omitted (previously lines 82 – 91). The general description of GUV preparation and the components used were moved to a new methods section (p.13 lines 350-360; GUV components and general protocol).***
- 2. As mentioned above, the entirety of section 2 of the results has been removed. The figure has been moved to the SI (now Supplementary Figure 7). The results of the experiment are briefly mentioned in the figure legend: ‘The reactions exhibited a 1.8- and 4.4-fold increase in fluorescence for 2 and 10 μ M, respectively.’ The results are reported in the main manuscript in the following section but before describing the autocatalytic self-replicating system. Specifically, we added (p.5-6 lines 135 – 145): ‘Typically, RNA ligase ribozymes require higher magnesium... indicating ribozyme catalysed ligation.’ The rest of the paragraph was modified to address the non-homogeneity of content exchange among protocells as requested by reviewer 1 comment #3 (p.6 lines 145 – 155). Finally, the experimental details were moved to a newly added supplementary methods section (see Supplementary Methods): ‘Fluorescent R3C ligase assay in GUVs for microscopy.’***

3. **We have omitted the detailed initial characterization of F1 and Hyper-A from the main manuscript (previously line 179 – 194). The text was replaced by a brief description of the system (lines 163 -165): ‘In this system, the ligase ribozyme (F1) replicates by joining two oligonucleotide substrates (Hyper-A and B, Supplementary Figure 1c). A reporter substrate (Cy5-A) was designed to carry an inactivating single point mutation (G38A) and a Cy5 fluorophore at the 5’-end (Supplementary Table 1).’**

We added ‘...characterizing and...’ at the start of the next paragraph (line 166) and cite the supplementary figures (line 166). The results are reported in the legend of Supplementary Figure 13. Finally, the description of the system and rationale behind the generation of the inactive reporter substrate have been moved to a new methods section (lines 460-469): ‘Generation of inactive R3C reporter substrate.’ A new supplementary methods section has been added to accommodate the methods for the experiments shown in the supplementary figures.

2.4 In a number of cases, the used experimental conditions are implied to be prebiotically plausible - with freeze-thaw cycles at -80°C. The claims of prebiotic plausibility need to be clarified and/or toned down. lines 59 and 94: The statement that 'less extreme freezing conditions' were used in the study compared to the liquid nitrogen freezing conditions in reference 22 is misleading when no additional detail is added - because the condition of -80°C stated in line 94 is not very prebiotically plausible either. Please add the detail of -80°C in line 59 or otherwise make the statement in line 59 more transparent. Similarly, the terms 'plausible geochemical environment' and 'geological scenario' in the discussion (lines 288 and 290) don't describe the used conditions.

We thank the reviewer for bringing this miscommunication to light. The term ‘less extreme’ was not meant to be understood in the frame of prebiotic plausibility, but rather in the context of the mechanism of content exchange that is observed for different temperature regimes : either content exchange mediated by vesicle fusion and fission (under more “extreme” freezing conditions employing centrifugation and the use of liquid nitrogen) or content exchange mediated by membrane permeabilization and diffusion observed under “less extreme conditions” (gravitation & metal block at -80 °C). Generally, homogeneous freezing of sample tubes or microtiter plates with pre-cooled metal blocks has been used in several other studies to generate eutectic water-ice, since the homogeneous cooling by the metal block minimizes unwanted super-cooling of μL samples (e.g. <https://doi.org/10.1038/nchem.2251> or <https://doi.org/10.1007/s00239-016-9729-9>). However, to avoid the notion that our conditions are more prebiotically plausible, we have removed an explicit comparison between both conditions. The paragraph now reads (lines 64-70): “For instance, repeated liquid nitrogen-based freeze-thaw cycles of pelleted lipid vesicles enable the coupling of fusion and fission of liposomes with RNA replication by a modern viral RNA replicase, and the distribution of RNAs within the vesicle population [28]. Previously, our groups investigated the effects of temperature cycling on giant unilamellar vesicles (GUVs), showing that freezing at -80 °C followed by passive thawing allowed content exchange between densely packed GUVs independently from fusion and fission [29].”

Next, the term ‘plausible geochemical environment’ has been omitted from the discussion. Similarly, the term geological scenario has been omitted. We describe what we postulate to be a geological scenario on early Earth that could have exhibited this type of phenomenon, modifying the first paragraph of the discussion to read (p.10 lines 262-275): “In this work, we explored how catalytic RNAs, including autocatalytic replicators, can propagate within a population of GUVs by exploiting transient, temperature-induced defects in the lipid bilayer. Following previous studies, we focused on

periodic freeze-thaw (FT) cycles as the main drivers of content exchange that may have also occurred on the early Earth, e.g., in the wake of day-night cycles in cold regions. The high vesicle concentrations required to minimize content loss require a scenario that leads to continuous vesicle enrichment. A suitable environment in this context could have been shallow surface streams flowing through porous rocks. Here, the pores can serve as traps in which the vesicles accumulate before being exposed to freezing and thawing conditions. The timescales and temperatures of plausible environmental temperature fluctuations clearly differ from the optimised lab conditions in our experiments (cooling at -80 °C to ensure homogeneous freezing and prevent super-cooled solutions followed by thawing at room temperature). Thus, future experiments may explore the mechanism established in this study under more plausible environmental temperature changes.”

Minor points

Line 39: The term 'readily available' may be an overstatement because the synthesis is not very efficient, and needs to proceed in a very different environment than the synthesis of all other compounds. I suggest replacing with 'prebiotically plausible' or similar.

We thank the reviewer for this suggestion. The sentence stating that they are readily available has been omitted, and the phrase has been changed to read (line 37-39): “While initial concepts of such vesicular protocells evolved around fatty acids, which are accessible via prebiotically plausible synthesis pathways⁹, their low stability and lack of tolerance towards physicochemical parameters such as pH and ionic strength made reconciling them with nucleic acid catalysis challenging^{10,11}”

line 85: Please define 'HH-FQ-sub' the first time this term is used.

The differences in the HH-substrates have been defined more clearly. Specifically, we state exactly what modifications each substrate has and for what experiments they were used. The description of the substrates now reads (lines 87-92): “...we opted for a well characterised hammerhead ribozyme (HH-min) that cleaves an RNA substrate strand, which either contains only a fluorescent dye (Cy5) at the 5'-end (named HH-Sub) or a fluorescent dye at the 5'-end (FAM) and a black hole quencher (BHQ1) at the 3'-end (named HH-FQ-Sub)³⁰ (Supplementary Figures 1a and 2). Cleavage of the RNA substrate can therefore be characterised both microscopically by the increase in fluorescence upon HH-FQ-Sub cleavage or by classical denaturing polyacrylamide gel electrophoresis (PAGE) upon HH-Sub cleavage.”

Lines 95 and following: Could a statement be made what fraction of cleaved Hh ribozyme substrate would correspond to the observed increase in fluorescence? While the PAGE analysis addresses that question it would be nice to have an estimate what fraction of substrate cleavage would be consistent with the observed increase in fluorescence.

We thank the referee for this insightful comment. We have now included a control experiments in the SI (Supplementary Figure 4) where HH-FQ-Sub was encapsulated alone or with HH-min, then was allowed to react and fluorescence intensity was measured. At the same concentration of substrate (2.5 μM), we observed an 8.3x increase in fluorescence intensity in the sample with HH-min compared to the sample without any ribozyme. By assuming that this corresponds to the total (100%) cleavage of the substrate, we can use the change in fluorescence to approximate what fraction of the HH-FQ-Sub is cleaved in the freeze-thaw experiment with HH-min (Figure 1b-c). Therefore, if the

total cleavage of 2.5 μ M substrate results in an 8.3x fluorescence intensity increase, we estimate the observed fluorescence increase in the freeze-thaw experiment (~6.7x) corresponds to approximately 80% cleavage. This corroborates the PAGE data regarding how much cleaved substrate the HH-min produces in a single freeze-thaw cycle.

We have added the short estimation to the text, updated the SI and remodelled the following sentences (lines 103 – 109): ‘To estimate the percentage of cleaved HH-FQ-Sub, we normalised the data obtained to a control experiment in which HH-min and HH-FQ-Sub were encapsulated in the same GUV population (Supplementary Figure 4). We estimate that approximately 80% of HH-FQ-Sub was cleaved after a single FT cycle. Finally, we collected the pre- and post-cycling GUV solutions and analysed the resulting RNA species by polyacrylamide gel electrophoresis (PAGE) to verify that FT-induced content exchange led to specific HH-sub cleavage.’

Figure 1: Please replace the acronym 'NC' with a more intuitive acronym (all other acronyms such as HH-mut and HH-min are intuitive). This will make it easier for the reader to understand the figure.

The acronym NC has been replaced with Neg Ctrl for perspicuity.

I suggest reminding the reader every now and then that "FT" means 'freeze-thaw'. The reader is not using this acronym on a daily basis and therefore needs more time to understand the sentences (it took me three readings of the sentence and one reference to line xx in order to understand line 195-197).

We thank the reviewer for this suggestion. We have added ‘freeze-thaw’ preceding the (FT) in a number of places throughout the manuscript to ameliorate the reading experience of the manuscript.

REVIEWERS' COMMENTS

Reviewer #1 (Remarks to the Author):

Comments on the rebuttal provided by Mutschler et al. (Jan 2023):

Overall, the scientific work under consideration has adequately addressed all our comments in the revised version of their manuscript. They have also repeated some experiments with better protocols and more robust results.

In lines 266-270: The authors mentioned the importance of high vesicle densities and justify how it might have occurred in natural settings on the prebiotic Earth. However, adding the references to the part pertaining to the porous rock scenario will be pertinent. Seems like Yandrapalli et.al.'s paper needs to be included?

Lines 390-393: The authors have elaborated on one critical point regarding different rates of heating and cooling. While the reasoning is clear, it still doesn't seem like a very prebiotically plausible scenario. However, it still is a nice proof of principle experiment.

Other than the aforementioned, the authors have, for the most part, defended their work and provided sufficient clarification for each of the comments that were raised as part of the previous round of review. I recommend that the manuscript be considered for publication in your journal.

Reviewer #2 (Remarks to the Author):

The authors have satisfactorily addressed all my comments. I support publication as is.

Response to Reviewer 1

1.1 Overall, the scientific work under consideration has adequately addressed all our comments in the revised version of their manuscript. They have also repeated some experiments with better protocols and more robust results.

We thank the reviewer very much for the positive assessment of our revised manuscript.

1.2 In lines 266-270: The authors mentioned the importance of high vesicle densities and justify how it might have occurred in natural settings on the prebiotic Earth. However, adding the references to the part pertaining to the porous rock scenario will be pertinent. Seems like Yandrapalli *et.al.*'s paper needs to be included?

We thank the reviewer for this insightful comment. We have now included the two references and added the following paragraph to the respective section (now lines 233-237):

"Studies have shown how micro-structured posts can trap and accumulate GUVs³⁴, supporting the claim that porous rocks could potentially concentrate protocells. Moreover, a model Hadean microcompartment mimicking hydrothermal porous rock with a gas-water interface was shown to promote the encapsulation of prebiotically relevant biomolecules as well as aggregate vesicles¹⁸."

1.3 Lines 390-393: The authors have elaborated on one critical point regarding different rates of heating and cooling. While the reasoning is clear, it still doesn't seem like a very prebiotically plausible scenario. However, it still is a nice proof of principle experiment.

We thank the reviewer for this assessment. While our experimental setup is of course a simplified and limited model system, the general concept of temperature-induced exchange of vesicle contents may be more general than initially expected. For example, we have preliminary unpublished data suggesting that exchange may also occur under less "harsh" conditions, i.e., less extreme temperature profiles that do not involve freezing or freezing at all. We will pursue this in future studies.

1.4 Other than the aforementioned, the authors have, for the most part, defended their work and provided sufficient clarification for each of the comments that were raised as part of the previous round of review. I recommend that the manuscript be considered for publication in your journal.

We thank the reviewer for this final evaluation. The critical feedback of all referees was very important for us and helped us to make this a much better manuscript.